# Uncovering the Biotechnological Importance of *Geotrichum candidum*

**DOI:** 10.3390/foods12061124

**Published:** 2023-03-07

**Authors:** Eleni Kamilari, Catherine Stanton, F. Jerry Reen, R. Paul Ross

**Affiliations:** 1APC Microbiome Ireland, University College Cork, T12 YT20 Cork, Ireland; 2School of Microbiology, University College Cork, T12 YT20 Cork, Ireland; 3Department of Biosciences, Teagasc Food Research Centre, Moorepark, Fermoy, P61 C996 Co. Cork, Ireland; 4Synthesis and Solid State Pharmaceutical Centre, University College Cork, T12 YT20 Cork, Ireland

**Keywords:** *Geotrichum candidum*, yeast, fungi, starter cultures, GRAS, biotechnology, lipases, cellulases, antimicrobial compounds, bioremediation

## Abstract

Fungi make a fundamental contribution to several biotechnological processes, including brewing, winemaking, and the production of enzymes, organic acids, alcohols, antibiotics, and pharmaceuticals. The present review explores the biotechnological importance of the filamentous yeast-like fungus *Geotrichum candidum*, a ubiquitous species known for its use as a starter in the dairy industry. To uncover *G. candidum*’s biotechnological role, we performed a search for related work through the scientific indexing internet services, Web of Science and Google Scholar. The following query was used: *Geotrichum candidum*, producing about 6500 scientific papers from 2017 to 2022. From these, approximately 150 that were associated with industrial applications of *G. candidum* were selected. Our analysis revealed that apart from its role as a starter in the dairy and brewing industries, this species has been administered as a probiotic nutritional supplement in fish, indicating improvements in developmental and immunological parameters. Strains of this species produce a plethora of biotechnologically important enzymes, including cellulases, β-glucanases, xylanases, lipases, proteases, and α-amylases. Moreover, strains that produce antimicrobial compounds and that are capable of bioremediation were identified. The findings of the present review demonstrate the importance of *G. candidum* for agrifood- and bio-industries and provide further insights into its potential future biotechnological roles.

## 1. Introduction

The yeast *Geotrichum candidum* (teleomorph = *Galactomyces candidus*) is a member of the basal family the Dipodascaceae, which belongs to the subdivision Saccharomycotina of the phylum Ascomycota within the kingdom of Fungi [1]. This ubiquitous, acid-tolerant species has been detected in several habitats, including soil, water, air, plants, and dairy products [2,3,4,5,6]. Additionally, it is a member of the commensal human skin, tracheobronchial tree, and gastrointestinal tract microbiota [7,8].

*G. candidum* strains present increased morphological and phenotypic diversity and characteristics similar to filamentous fungi [3,9]. Therefore, the species was initially classified as yeast. However, later analysis contributed to its reclassification as a filamentous, yeast-like fungus or a mold [10,11]. Although *G. candidum* is self-fertile, a sequence-based analysis indicated that *Galactomyces candidus*, formerly believed to comprise a distinct taxon, is its sexual state [4,12,13]. Morel and coworkers [10] identified a gene (GECA02s02545g) coding for a protein named MATA, which, based on comparative genomics with other yeasts and fungal species, is responsible for *G. candidum* sexual state. Kurtzman and Fell [3] proposed that *G. candidum* strains can be distinguished from the *Galactomyces* species due to their weak or positive metabolism of α-ketoglutaric acid and malic acid, respectively.

The species is also known to form colonies on the surfaces of soft cheeses, creating a creamy-white color on brie and camembert rind, and growing on cheese boxes made of wood [14,15]. Furthermore, it has been widely applied as a starter for the process of ripening in mature cheeses [16], as well as in malting and brewing processes [17], to diminish fungal contamination and to improve the products’ quality. Furthermore, strains of the species are considered promising candidates for industrial administration, due to the release of enzymes, such as cellulases, α-amylases, proteases, lipases, β-glucanases, xylanases, and phytases [15,18,19,20]. Additionally, strains of these species indicated biofilm-forming capacity [21].

So far, there is no connection between the consumption of food products containing *G. candidum* and any foodborne disease. It is also not part of any list of biological pathogens of the French regulation regarding biological safety levels or the Advisory Committee on Dangerous Pathogens (2004) [2]. Additionally, the European Food and Feed Cultures Association (EFFCA) and the International Dairy Federation (IDF) list *G. candidum* among the microorganisms with a recognized history of safe administration in fermented foods [22]. However, the United States Food and Drug Administration (FDA or US FDA) has neither provided a regulation nor evaluated a generally recognized as safe (GRAS) notice for the use of *G. candidum* in human food (https://www.fda.gov/food/generally-recognized-safe-gras/microorganisms-microbial-derived-ingredients-used-food-partial-list (accessed on 1 April 2018)). However, based on their condition of use, strains of *G. candidum* and/or substances in a conventional food derived from *G. candidum* might have the potential to obtain a GRAS status. The present review article aims to reveal the current applications of *G. candidum* and shed light on potential future biotechnological applications of this yeast-like fungus.

## 2. Administration as a Starter Culture

### 2.1. Cheeses

*G. candidum* is present in raw milk at low concentrations (<10^2^ colony-forming units (CFU)/mL) [23,24]. Therefore, it is detected in several dairy products, including fermented milk products (e.g., viili, kefir) and soft and ripened cheeses [3,25,26,27,28]. Its capacity to digest lactic acid, producing alkaline products, favors its growth during the process of ripening [5]. This deacidification performed on the cheese surface promotes the growth of acid-sensitive bacteria [29,30,31,32]. Investigation of the species growth dynamics in the presence of the commercial lactic acid bacteria (LAB) culture DVS^®^ FRESCO^®^ 1000NG indicated that *G. candidum* can mutually coexist and grow with LAB, the main dominant bacteria found in cheeses [27]. Its enhanced proteolytic and lipolytic activity influences the development of flavors and aromas in cheese [33]. Indeed, the addition of *G. candidum* in combination with LAB as starters in a soft Chinese soy cheese contributed to increased protein and fat degradation, leading to elevation of the pH value, a reduction in the unwanted beany taste, and a reduction in hardness during ripening [34]. Additionally, its administration as a starter culture in combination with LAB (*Lactococcus lactis* subsp. *lactis* biovar *diacetylactis* and *Leuconostoc mesenteroides* subsp. *cremoris*), and variants of the mold *Penicillium*, such as *Penicillium camemberti*, were considered responsible for the specific flavors and aromas of camembert cheese, a soft cheese with increased humidity that is covered with a white rind [35,36]. The deamination reactions performed by *G. candidum*, which lead to the production of aldehydes, ammonia, and organic acids, influence the cheese’s sensorial characteristics [3,37,38]. In general, *G. candidum*-associated sensorial characteristics include mildly cheesy, sweaty, moldy, yeasty, fermented, acidic, putrid, cidery, musty, and fruity flavors, and lightly velvety or fluffy appearance [3,39].

Whole genome sequencing analysis of *G. candidum* strain CLIB 918 (ATCC 204307) identified the presence of genes favoring its growth in the cheese microenvironment, such as genes coding for lipases, and production of volatile sulfur compounds responsible for specific aromas [18]. In agreement, a comparison of the metabolic capability of *G. candidum* CLIB 918 and *G. candidum* strain Gc203, a stronger sulfur compound producer, revealed that the majority of the produced organic compounds were generated from catabolism of branched-chain amino acids, sulfur-containing amino acids, and fatty acids [40]. Their ability to produce these volatile compounds was negatively associated with their capacity to store amino acids inside the cell. The same researchers indicated that the production of hypotaurine, taurine, and glutathione, and the amount of volatile fatty aldehyde production inside the cell was diminished in the induction of oxidative stress. Furthermore, the production of aromatic compounds by the end of ripening at the cheese surface was regulated by oxygen and iron.

Perkins and coworkers [41] analyzed the carbon assimilation profile of eleven *G. candidum* and *Galactomyces* spp. strains to reveal a weak or negative citrate utilization ability, suggesting that the metabolites produced through citrate utilization are not critical contributors to surface-ripened cheese flavors [5,30,42]. By contrast, all analyzed strains were able to perform transamination of L-methionine, producing glutamic acid, which together with an amino-acid acceptor, such as α-ketoglutaric acid, constitutes a major contributor to ammonia and other flavor compounds produced in ripened cheeses [3,30,43,44,45]. Moreover, the analysis revealed that the metabolic pathways contributing to cheese sensorial characteristics are strain-dependent [41]. 

Castellote and coworkers [29] analyzed the metabolic activity and the physiology of *G. candidum* during the ripening of Reblochon-type cheeses, by quantifying mRNA transcripts. At the beginning of ripening, overexpression was observed in genes associated with acetate, lactate, and ethanol catabolism. The majority of genes that were negatively affected during ripening involved genes related to the organization of the cell wall, protein synthesis, vesicular-based transport, and cytoskeleton components. On the contrary, genes implicated in the transportation and utilization of amino acids, such as glutamate, and transcripts stimulating autophagy and affecting lifespan, were overexpressed. The expression of these genes possibly indicates the necessity for a metabolic shift to amino acid catabolism associated with nutrient availability and starvation during cheese ripening.

Apart from their contribution to cheese sensorial characteristics, some strains were reported to display increased antimicrobial activity against spoilage microbes, such as *Mucor* spp., *Aspergillus ochraceus*, and pathogens, such as *Listeria monocytogenes*, *Staphylococcus aureus*, *Escherichia coli*, and *Bacillus cereus* [3,14,46]. It is noteworthy that *G. candidum* is considered a spoilage organism in specific dairy products, including cream, butter, and cottage cheese. Its excessive growth may cause off-flavor development and the appearance of defects in cheese, including toad skin or slippery rind [10]. Additionally, the species’ ability to grow on any surface attached to food products raises concerns about possible economic defects for food producers [47].

Luo and coworkers [48] identified the *G. candidum* gene encoding the enzyme Δ12 fatty acid desaturase, which catalyzes the desaturation of oleic acid (OA) and linoleic acid (LA) to produce LA and α-linolenic acid (ALA), respectively. Molecular cloning of the gene named GcFADS12 in *Saccharomyces cerevisiae* contributed to increased production of LA and ALA. The researchers considered the potential administration of the enzyme in soft cheeses with a white rind, characterized by the absence of essential fatty acids. Shi and coworkers [49] identified allosteric and active sites at the catalytic domain of the enzyme.

### 2.2. Cheeses Analogs

Łopusiewicz and coworkers [50] used LAB and the fungi *P. camemberti* and *G. candidum* as starter cultures for the production of flaxseed oil cake, an analog of Camembert cheese. Flaxseed oil cake is considered a “superfood” due to its high protein content and the presence of bioactive compounds, including ALA, lignans and phenolic compounds, antioxidant compounds, as well as dietary fibers [51,52]. The study revealed that the bioactivity and the product’s physicochemical characteristics were influenced by the administration of *G. candidum* [50]. Specifically, combinatory administration of *P. camemberti* and *G. candidum* (PC + GC), resulted in higher fungal viability compared to administration of *P. camemberti* only (PC). This was possibly due to elevated secretion of hydrolytic enzymes, leading to more successful substrate utilization and biotransformation productivity. For instance, *G. candidum* is capable of catabolizing bitter peptides produced by *P. camemberti* [50]. In fact, a significant increase in the total free amino acid levels in sample PC + GC was observed during the ripening. The increased proteolysis affected the texture of the product, resulting in reduced hardness and chewiness in the PC + GC sample. Due to amino acid deamination and the release of ammonia, the pH value increased significantly and the titrable acidity reduced significantly in the PC + GC group. Furthermore, an effect in the total polyphenolic content and total flavonoid content was indicated, possibly due to the release of antioxidant compounds as a result of *G. candidum*-associated lipolytic activity. 

### 2.3. Malting

*G. candidum* has also been proposed as a starter culture in malting, to decrease the presence of fungal contaminants and to enhance the qualitative characteristics of malt [11,53,54]. The ability of *G. candidum* to produce extracellular enzymes, such as cellulases, polygalacturonases, xylanases, and glucanases, allows the species to successfully colonize barley grain [53]. Boivin and Malanda and Linko and coworkers [11,54], indicated that the species administration resulted in improvement of malt extract, increased amount of soluble nitrogen, grain fragility, Kolbach index, and reductions in beta-glucan levels and viscosity. Additionally, the release of β-1,3-glucanase and chitinases by *G. candidum* results in a reduction in mold contaminants, as glucan and chitin form the ingredients of the mold cell wall [53,55]. Recently, Kawtharani and coworkers [56] indicated that the production of phenyllactic acid by *G. candidum* decreases the growth of *Fusarium sporotrichioides* and *Fusarium langsethiae* and the production of T-2 toxin during malting. T-2 toxin belongs to the type A trichothecenes and is considered the most cytotoxic among the members of this group, negatively affecting cellular metabolism and causing carcinogenicity in some affected animals [57].

## 3. Administration as a Probiotic Strain 

Following administration to the rohu fish, *Labeo rohita*, the effects of the probiotic strain *G. candidum* QAUGC01 were evaluated in terms of growth rate, intestinal enzymatic activity, and immune system function [58,59,60,61]. Ibrar and coworkers [58] studied the impact of *G. candidum* QAUGC01 administration in the rearing water of *L. rohita* at the postlarvae stage, whereas Amir and coworkers [59] analyzed the effects of the commercial strain administration as a dietary supplement in its free and encapsulated form to fingerlings of *L. rohita*. Both studies indicated significantly elevated percentages of survival; enhanced growth parameters, such as weight gain and development rate; improved intestinal amylase, protease, and cellulase activity; and advanced immune activity. Additionally, significantly reduced mortality following challenge with *Staphylococcus aureus*, improved hemato-immunological indices and increased gene expression of the heat shock protein HSP 70 in the intestine, muscles, and liver, and reductions in triglyceride, total cholesterol, and serum alanine aminotransferase and aspartate aminotransferase function were observed. The effects were significantly greater in the group fed the encapsulated strain compared to those supplemented with the unencapsulated strain [59]. The same team compared the effects of the probiotics (a) *G. candidum*; (b) *B. cereus*; (c) *G. candidum* and *B. cereus* [61] and (a) *Enterococcus faecium*; and (b) *E. faecium* and *G. candidum*; [60] as nutritional supplements to *L. rohita* compared with no probiotic administration (control). Probiotic supplementation with *G. candidum* only resulted in significantly improved growth parameters, intestinal enzymatic activities, and survival rates, especially after challenging with *Aeromonas hydrophila*, compared to the control, *B. cereus* only, and as a co-culture with *B. cereus* [61]. Similarly, these parameters were improved after combinatory administration of *G. candidum* and *E. faecium* compared to the control and the *E. faecium*-supplemented diet [60]. Similar research evaluated the possible probiotic effects of *G. candidum* administration to Gibel carp CAS Ⅲ (*Carassius auratus gibelio*) [62]. In agreement with the experiments on *L. rohita*, *G. candidum* improved growth, intestinal α-amylase activity, and immune-related gene expression and response compared to the control.

## 4. Antimicrobial Activity of *G. candidum*

*G. candidum* has been reported to produce antimicrobial compounds, including indoleacetic acid, phenyllactic acid, phenylethyl alcohol, and their derivatives [46,56,63] in addition to antimicrobial peptides. Phenyllactic acid has a broad-spectrum antimicrobial activity against bacteria, yeasts, and molds, targeting the microbial cell wall. Phenylethyl alcohol is a bacteriostatic agent, which in low concentrations reversibly inhibits the synthesis of deoxyribonucleic acid, whereas in higher concentrations (90 to 180 mM) it affects the plasma membrane, as well as sugar and amino acid transport systems. Indoleacetic acid and phenyllactic acid prevent the growth of *L. monocytogenes*, causing behavioral and structural alterations, whereas phenylethyl alcohol causes disruption of the cell membrane and inhibition of protein synthesis [46]. Aromatic amino acids were identified to influence the secretion of antimicrobial compounds in some *G. candidum* isolates [14]. Specifically, phenyllactic acid is derived from phenylpyruvic acid following transamination from phenylalanine. Additionally, phenylethyl alcohol is produced following decarboxylation and deamination of phenylalanine. The endophytic *G. candidum* PF005 isolated from the fruit of eggplant (*Solanum melongena*) was reported to secrete volatile organic compounds (VOCs), such as ethyl 3-methylbutanoate, 3-methyl-1-butanol, 2-phenylethanol, naphthalene, isopentyl acetate, and isobutyl acetate with increased antifungal activity against the phytopathogen *Rhizoctonia solani* [64]. Moderate antifungal activity was also observed against other fungi, such as *Fusarium oxysporum*, *Pseudocercospora* sp., and *Cercospora* sp. [64]. The production of VOCs was enhanced by specific alcohol and ester volatile precursors, and exogenous naphthalene addition to the growth medium. The same team isolated and characterized the GcAAT gene that encodes the alcohol acetyltransferase GcAAT, which was found to be responsible for the secreted *G. candidum* PF005 antifungal VOCs [65]. Structural analysis identified the catalytic domain of the enzyme inside a tunnel that contained specific binding sites for three alcohols (i.e., isobutyl alcohol, isoamyl alcohol, and 2-phenylethanol) and the co-substrate acetyl-CoA.

Omeike and coworkers [63] identified a novel antimicrobial peptide (AMP) produced by the strain *G. candidum* OMON-1 isolated from pharmaceutical wastes. The AMP was purified using reverse-phase high-performance liquid chromatography (RP-HPLC). The purified peptide (GP-2B) was identified using liquid chromatography-mass spectrometry (LC–MS) and MALDI-TOF tandem mass spectrometry (MS/MS). The pure, positively charged GP-2B had a molecular mass of 409.23 atomic mass units (amu) and contained a pattern of carboxymethylcystyl–asparagyl–aspartate amino acids in its sequence. This peptide indicated bacteriostatic effects against *S. aureus* and enterococcal strains.

*G. candidum* LG-8 was proposed as a potential probiotic candidate to control the growth of pathogenic *Pseudomonas aeruginosa* in food products and immunodeficient patients due to its ability to adhere to/trap the pathogen. Investigation of the adhesion ability of *P. aeruginosa* PAO1 to *G. candidum* LG-8 indicated that both live and dead yeast cells were able to adhere to PAO1 [66]. This ability was restricted to the presence of pH values between 2.0 and 9.0 and bile salts concentrations of more than 0.5%. Within 4 h, about a hundredfold PAO1 were adhered. Additionally, the strain *G. candidum* MK880487 was observed to produce a glycolipid biosurfactant that was active against the phytopathogenic fungus *Macrophomina phaseolina* [67]. GC-MS analysis of the crude extract showed that the hydrophobic content, apart from fatty acids, consisted of Lucenin 2 and luteolin-6,8-di-C-glycoside. The ability of two *G. candidum* and one *Galactomyces pseudocandidum* strains, isolated from rhizosphere soil in Egypt, to produce significant amounts of biosurfactants was confirmed by Eldin and coworkers [68].

Furthermore, Shalaby and coworkers [69] created a biologically synthesized nanoparticle, [*G. candidum*/FeO + P_2_O_5_]^NC^, with increased inhibitory activity against *L. monocytogenes*, *P. aeruginosa*, *B. cereus*, *Salmonella typhi*, *E. coli*, *Enterococcus faecalis*, and *Candida albicans*. Using the agar well diffusion method, [*G. candidum*/FeO + P_2_O_5_]^NC^ in 0.2 g/L concentrations created inhibition zones with diameters of 15 to 20 mm. A comparison of its activity with synthetic antibiotics and anti-fungal compounds revealed that [*G. candidum*/FeO + P_2_O_5_]^NC^ decreased the total bacterial count and the total fungal count in polluted water by 90.5% and 99.4%, respectively, whereas the antibiotic amoxicillin reduced them by 54.8% and 97.7%, respectively, and the anti-fungal nystatin by 84.9% and 98.57%, respectively. The authors suggested that the production of reactive oxygen species (ROS), such as H_2_O_2_, which destroys the bacterial cell wall, might be responsible for the increased inhibitory activity of the doped [FeO + P_2_O_5_]^NC^ [70,71].

## 5. Production of Enzymes with Industrial Interest 

### 5.1. Lytic Polysaccharide Monooxygenases (LPMOs)

Increasing demands of transportation fuel consumption (i.e., oil and biofuels), have prompted industries to invest in potentially renewable sources of energy production, such as plant biomass [72]. The main components of plant cell walls are cellulose, some non-cellulosic polysaccharides, and lignin. Plant biomass-derived lignocellulosic residues are considered great sources of fermentable sugars, the processing of which may lead to the production of renewable liquid transport fuels [73,74,75,76]. Morel and coworkers [10] identified the presence of genes encoding lignocellulolytic enzymes in the genome of *G. candidum* strain CLIB 918 (ATCC 204307). These enzymes included lytic polysaccharide monooxygenases (LPMOs) of the AA9 family, GH45 endoglucanases, which are strong oxidative enzymes that are important for the degradation of cellulose and hemicelluloses, such as xyloglucan and glucomannan [77], and endo-polygalacturonases. Similarly, *G. candidum* 3C was found to encode functional lytic LPMOs of the AA9 family [78,79]. LPMOs produced by *G. candidum* are promising candidates for enzymatic cocktails applied in biorefineries enrichment. Moreover, a cellulase isolated from *G. candidum* GAD1 was found to be promising for the degradation of carboxymethylcellulose salt from agricultural waste (rice straw), the fermentation of which resulted in bioethanol production [80].

Due to their improved catalytic activity against cellulose and hemicelluloses, some *G. candidum* strains were used for filter paper and cotton degradation. For instance, *G. candidum* strain 3C was discovered to encode a glycoside hydrolase (GH) of the family 7 cellobiohydrolases (CBHs), named Cel7A [19]. This strain’s cellulase complex exhibited more effective activity than that of *Hypocrea jecorina*, the most commonly used species for cellulase production [81]. An enzymatic cocktail from *G. candidum* strain 3C named ‘Cellokandin G10x’ has been applied for industrial pulp and wastepaper utilization [19]. Similarly, enzymatic isolates from *G. candidum* strain Dec 1 were found to improve kraft pulp bleaching [82]. Noteworthy, ITS, 18S rDNA, 28S rDNA, and RPB2 gene sequence comparisons and multiple sequencing analysis of *G. candidum* strain 3C indicated that the strain should be included within the genus *Scytalidium* (Pezizomycotina, Leotiomycetes) and renamed *Scytalidium candidum* 3C comb. nov. [83].

### 5.2. Lipases

*G. candidum* can produce extracellular lipases, especially when cultured in the presence of an inducer, such as triglycerides or olive oil, in the culture medium [84,85,86,87,88]. Lipases have numerous industrial applications, including the manufacture of enantiomerically pure pharmaceuticals, cosmetics, agrochemicals, surfactants, biolubricants, construction and destruction of biopolymers, waste-water-treatment, influence on food products’ sensorial characteristics, detergent industries, etc. [89,90,91,92]. Ferreira and coworkers [93] used a lipase secreted by *G. candidum* (GCL-I) to produce free fatty acids from olive, palm kernel, and cottonseed oils. Glycerol and free fatty acids are used by oleochemical industries to produce several products, including personal care products, such as shampoos, coatings, adhesives, surfactants, fatty alcohols, and lubricating oils [94,95,96]. *G. candidum* (ATCC 34614) was found to produce four lipases with different substrate specificities [97]. Specifically, lipases I and II indicated non-regional specificity with triolein, while lipases III and IV catalyzed the hydrolysis of triolein oleoyl esters at the sn-2 position with greater specificity compared to the sn-1(3) position, indicating an sn-2-regioselectivity. Laguerre and coworkers [98] indicated that the lipases isolated from *G. candidum* NRRL Y-552 were able to hydrolyze triolein and six edible oils, producing high amounts of diacylglycerol (DAG)-1,3 and lower amounts of DAG-1,2(2,3). The reduction in DAG-1,3 in oils with increased DAG-1,3 concentration is of great importance for the production of foods containing nutraceutical diacylglycerols able to reduce the levels of triacylglycerol in the plasma. Moreover, *G. candidum* lipase (LGC-I) was administered to produce decyl oleate ester from rice husks (phenyl-silica), which were chemically modified [99]. Furthermore, GCL-I and -II were applied for the hydrolysis or ethanolysis of Crambe and Camelina oils, producing high concentrations of erucic acid and gondoic acid [100]. 

To be applied as a biocatalyst, the lipases produced during fermentation by *G. candidum* need to be immobilized. Ferreira and coworkers [101] analyzed different approaches for purified lipase immobilization, to reveal that the most efficient protocol was ionic adsorption on MANAE-agarose. The molecular masses of the currently identified and purified lipases produced by *Geotrichum* sp. range from 32 kDa to 75 kDa [18,90,102,103,104,105,106]. Moreover, administration of endo-β-N-acetylglucosaminidase showed that these lipases are glycosylated, and beyond their high homology, they have different substrate specificity [107]. Furthermore, chemically modified to contain more amino groups, surface lipase produced by *G. candidum* (GCL) indicated faster and easiest immobilization on carboxymethyl and sulfopropyl agarose-based supports [107,108]. Administration of the ionic derivatives into fish oil resulted in increased hydrolysis and production of a high yield of Omega-3 polyunsaturated fatty acids (6.65 U and 7.85 U per gram of support of carboxymethyl derivative and sulfopropyl derivative, respectively) [107].

### 5.3. Alkaline Proteases

Investigation into the production of proteases from 30 *G. candidum* strains revealed that the proteolytic activity differs among the different strains [109]. More recently, 12 strains were tested for their ability to secrete proteases, from which one strain, *G. candidum* GCQAU01 isolated from the fermented milk product Dahiwas, was found to produce an ideal protease for industrial application [110]. Specifically, the identified serine-type protease indicated thermostability, had stable activity at temperatures ranging from 25 to 45 °C and pH 8–9, possessed increased hydrolytic activity against casein and bovine serum albumin (BSA), and its activity was prevented by PMSF (7.5%). Alkaline proteases are of great biotechnological importance due to their numerous applications in food, pharmaceutical and tannery industries, silver recovery, detergents and waste treatment, and amino acid resolution [111].

### 5.4. Pectinases

Pectinases produced by *Geotrichum* spp., such as *G. candidum* AA15, are considered good candidates to be applied by the fruit juice industry for clarification of juices [112,113]. The presence of pectin and several other components of fruits makes the produced juice cloudy [114], degrading its qualitative characteristics and affecting consumers’ preferences. Additionally, increased levels of pectin lead to an unwanted colloid texture formation [114]. The application of filtration can reduce the presence of pectin [115]. However, the increased presence of fiber-like molecules in pectin’s structure makes filtration inefficient for eliminating its existence in juice. Therefore, the addition of pectinolytic enzymes before the filtration process has been applied for the depectinization of several juices [116]. As a result, the process efficiency is increased [117]. The majority of pectinases are isolated from filamentous fungi, including *Aspergillus niger* [118]. However, the administration of pectinases from yeasts, such as *Geotrichum*, offers the advantages that: (a) some species are considered as GRAS [119]; (b) they produce an efficient type of pectinase for juice clarification [113]; and (c) they produce considerable amounts of pectinases in a shorter fermentation period compared to filamentous fungi [120]. Ahmed and Sohail [112] applied a response surface approach to reveal that the pectinase isolated from *G. candidum* AA15 was efficient for orange juice clarification, with the highest enzymatic activity in incubation time of 25 min at 35 °C, in pH 5. To increase the *G. candidum* AA15 pectinase yield, the strain was immobilized using corncob [113]. Additionally, simple sugars, such as xylose, galacturonic acid, galactose, and pectin, but not glucose, positively affected pectinase production in immobilized yeast cells [121].

### 5.5. Aldehyde Dehydrogenases, Glutamate Dehydrogenases, and Baeyer–Villiger Monooxygenases 

Several enzymes associated with oxidation reactions have been applied by pharmaceutical and chemical industries for the oxidation of alcohols, sulfides, aldehydes, Baeyer–Villiger oxidation, and hydroxylation [122,123,124,125,126,127]. *G. candidum* was discovered to produce dehydrogenases with broad applications in organic synthesis [125,128,129,130,131,132,133,134]. For instance, *G. candidum* NBRC 4597 (GcAPRD) was discovered to produce a novel alcohol dehydrogenase (ALDH), acetophenone reductase, an enzyme with broad spectrum activity regarding the oxidation of aldehydes to carboxylic acids and selective activity for the oxidation of dialdehydes to aldehydic acids [129,135,136]. This process has important agrochemical and pharmaceutical applications, but there are still limitations on its use due to its limited recyclability [137]. The specific ALDH was found to have broad-spectrum activity, thermostability, and resistance to non-aqueous solvents [133,134,138,139,140,141]. The application of an organic–inorganic nanocrystal formation method successfully immobilized the isolated GcALDH nanocrystal, which retained its enzymatic activity and proved more thermostable compared to the free GcALDH [142]. The same team used graphene oxide and reduced graphene oxide, which are chemically produced oxidized forms of graphene, to immobilize GcAPRD, enabling its recycling [143]. As a result, a great number of ketones, such as the aliphatic ketone 3-hexanone, were decreased with 99% efficacy. 

*G. candidum* S12 was shown to produce a novel glutamate dehydrogenase, which was highly active against glutamate, hexanol, α-ketoglutarate, and isoamyl alcohol (Km values of 41.74, 4.01, 20.37, and 19.37 mM, respectively) [144]. The catalytic activity against hexanol was enhanced by the addition of ADP, K^+^, Fe^2+^, and Zn^2+^ and reduced by EDTA, Mn^2+^, Pb^2+^, ATP, and DTT.

Furthermore, immobilization of the whole *G. candidum* CCT 1205 cell on functionalized silica resulted in ε-caprolactone creation, an advanced polymer with several biomedical applications [145]. E-caprolactone was also produced by whole *G. candidum* CCT 1205 cells using cyclohexenone, cyclohexanone, and cyclohexanol as substrates [146].

## 6. Other Important Biotechnological Applications

### 6.1. Oleaginicity

*G. candidum* NBT-1 isolated from rotten apples indicated an increased ability to accumulate single-cell oils (SCO) [147]. This ability is called oleaginicity (storage excess of 20% of yeast dry weight per cell) and emerges when there has been lack of essential nutrients (phosphorous, nitrogen, or rarely sulfur) and increased availability of carbon [148]. The produced lipids contained a high concentration of the rare medium-chain fatty acid (MCFA) caprylic acid, which has several medical and nutritional applications [149]. Since MCFA are an important nutritional source of rapid energy, they are administered in reduced-lipid-content food alternatives for the control of obesity [150]. Additionally, they have a key role in the treatment of epilepsy, carnitine deficiency, and hyperalimentation, in decreasing serum cholesterol, and in infant formula components [149,151,152]. Moreover, their increased similarity to plant oils makes them good candidates for raw materials for biofuel production. 

### 6.2. Heavy Metal Removal from the Ecosystem

Contamination of water by heavy metals, such as lead, constitutes a severe threat to ecosystems [153]. Lead is considered among the most hazardous pollutants and is included by the US Environmental Protection Agency (EPA) in the list of priority pollutants [154]. The presence of Pb^2+^ ions in the aquatic environment threatens the survival of aquatic life [155] and poses a serious risk to human health [156]. Even at low concentrations (more than 10 μg/L in drinking water according to the World Health Organization (WHO)), Pb^2+^ ions may cause hepatic damage and reproductive system dysfunction [157,158]. Bioremediation, the process that takes advantage of the heavy-metal biosorption potential of microorganisms, has been recently applied as a novel strategy to remove these pollutants from the environment [159]. The strain *Geotrichum* sp. CS-67, isolated from surface sediments in China, demonstrated the ability to survive in and accumulate high levels of Cu^2+^, Zn^2+^, and Ni^2+^ [160]. Ni^2+^ and Zn^2+^ were transported actively inside the cell, whereas Cu^2+^ was assimilated in the cell wall. Gene expression analysis revealed that the genes SED1 and GDI1 were upregulated during Ni^2+^ and Zn^2+^ exposure, whereas the expression of the gene ZRT1 was suppressed. 

The *G. candidum* strain LG-8, isolated from spontaneous Tibet kefir in China, demonstrated the ability to eliminate Pb^2+^ ions from water [161]. The strain was administered at low concentrations and was able to eradicate a total of 325.68 mg lead/g of dry biomass, as a result of adsorption to the cell wall. The use of bioagents, such as the yeast *G. candidum* strain LG-8, may provide several advantages over other heavy-metal detoxification technologies, including solvent extraction [162], ion exchange [163], iron oxide-doped sol-gel organic-inorganic hybrid nanocomposite [164], dried water hyacinth [165] and chemical [163,166]. Apart from being less economical, these strategies may not be very effective in dealing with low concentrations of heavy metals in water (1 to 100 mg/L) [163] and may produce undesired secondary metabolites [162]. 

The synthesis of nanoparticles using fungi is called myco-synthesis of nanoparticles, whereas the utility of fungal components and metabolites is named myco-nanotechnology [69]. These microbial nanocomposites are combined with multiplex materials, including metals, inorganic ceramics, and polymers, which have a diameter lower than 100 nm [167]. The inorganic compounds increase the strengthening properties of the structure, whereas the organic compounds perform the biocompatibility and biodegradability function. Their combination is performed using several processing techniques, such as solid-state mixing, solvent casting, in situ polymerization, melt blending and melt extrusion [167]. The development of bio-nanoparticles from components of the fungal cell wall could provide an effective practice for the absorption of metal ions. Specifically, the ability of *G. candidum* strain LG-8 to tolerate and remove high concentrations of lead is based on the strain’s cell wall components, including amide and carboxyl groups and phosphate. These components can bind lead, producing micro/nanoparticles of pyromorphite [Pb_5_(PO_4_)_3_Cl], a lead mineral that is very insoluble, with phosphorus and chlorine, offering a novel, eco-friendly approach to lead removal from the aquatic environment. The hybrid bio-nanocomposite [*G. candidum*/FeO + P_2_O_5_]^NC^ created by Shalaby and coworkers [69] was tested for the ability to remove heavy metals from a polluted aquatic environment. The doped [FeO + P_2_O_5_]^NC^ was capable of absorbing 91.4% of the lead. In addition, it revealed low removal capabilities for other metals, such as nickel, chromium, cadmium, and zinc (ranging from 5.7% to 23.4%).

### 6.3. Ability to Degrade Electronic and Electrical Waste (EEW)

*G. candidum* isolated from soil polluted by electronic wastes indicated the ability for biodegradation of rechargeable batteries and printed circuit boards [168]. The colorimetric method following incubation of the strain with keratine powder identified the production of keratinolytic enzymes able to degrade 23% of battery waste and 71% of circuit boards after 30 days of incubation. The administration of an environmentally friendly approach to recycling EEW wastes is of great importance, considering that their numbers are steadily increasing. Specifically, in 2019, their production reached 53.6 million metric tons (Mt) globally, indicating a 9.2 Mt increase compared to 2014, and it is expected to reach 74.7 Mt by 2030 [169].

### 6.4. Ability to Degrade Organic Waste

*G. candidum* species were detected in industrial wastewater and sewage [170], and indicated the ability to degrade several forms of organic waste, including phenolic compounds [171,172] and glycerol trinitrate [173]. Moreover, culturing the species in a medium with 0.5% concentration of ethoxylated oleyl–cetyl alcohol caused an increase in the production of alkaline protease by 55.84%, indicating its possible administration in bioremediation and detergent industries [174].

### 6.5. Decolorization of Textile Effluent

*G. candidum* was reported to be able to decolorize various dyes, including different azo and anthraquinone dyes [175]. Additionally, it has been proposed as a possible candidate for the decolorization of pollutant wastewater released by the textile industry [82,176,177]. Discarded dyes pose a major threat to the marine ecosystem and terrestrial organisms [178]. Traditional treatment procedures combined with chemical- and physical-based treatments are generally expensive, not very effective, and produce a great amount of waste, including sludge and gases that are difficult to dispose of [179]. The enzymes produced by some *G. candidum* strains provide the possibility of an eco-friendly and cost-effective strategy for the degradation of released synthetic dyes and molasses. Rajhans and coworkers [176] immobilized *G. candidum* using coconut fibers as a support material, and revealed that the decolorization activity was due to the activity of the ligninolytic enzyme laccase.

### 6.6. Encapsulation of Curcumin

Successful curcumin encapsulation via a vacuum infusion process using cell wall ingredients from the arthrospore of the probiotic *G. candidum* LG-8 and beta-1,4-glucan as the main components was recently reported [180]. Curcumin is a polyphenolic compound used extensively as a food additive, and is produced by the plant genus *Curcuma*, a member of the *Zingiberaceae* family [181]. Recent studies revealed that curcumin possesses antibacterial, anti-fungal, anti-viral, anti-inflammatory, anti-cancer, antioxidant, anti-diabetic, and hypolipidemic activities [182]. The method of encapsulation improves the bioaccessibility and stability of curcumin. The advantages of using *G. candidum* LG-8 include its biosafety and the massive and rapid production of arthrospores on several media without the need for specialized equipment [161]. 

### 6.7. Degradation of Biogenic Amines

Biogenic amines (BAs) involve a category of nitrogen-containing organic compounds, which are created by decarboxylation of amino acids and amination of ketones and aldehydes [183]. The most commonly detected BAs in fermented foods or beverages include tyramine, histamine, cadaverine, putrescine, β-phenethylamine, tryptamine, spermine, and spermidine. Their accumulation in food is associated with spoilage [184]. Therefore, limiting their presence is important for the food industry. While *G. candidum* has been shown to degrade tyramine and histamine from sausages [185], two strains of *G. candidum* were also identified as possible histamine-forming species in Cabrales and Pecorino Crotonese cheese [186,187].

## 7. Current Genomic Information

The different strains of *G. candidum* exhibit significant phenotypic and metabolic diversity [188,189,190,191]. Therefore, their identification and characterization using traditional microbiological approaches are difficult [188]. Additionally, they display significant diversity in their genome sizes and the sequences of their housekeeping genes [18,192,193]. Alper and coworkers [13] evaluated the rDNA sequence diversity of 62 *G. candidum* isolates to reveal increased polymorphism, based on which 32 different sequences were identified. A follow-up study evaluated the sequence diversity of 40 *G. candidum* isolates based on six housekeeping genes, including alanyl-tRNA synthetase (ALA1), phosphoglucoisomerase (PGI1), pyruvate kinase (CDC19), glutaminyl-tRNA synthase (GLN4), acetyl-coA acetyltransferase (ERG10), and phosphoglucomutase (PGM2), using multilocus sequence typing (MLST) and ITS1-5.8S-ITS2 region analyses [187]. The analysis identified that *G. candidum* isolates are separated into phylogenetically distinct clusters, with some populations being associated with cheese-making. In agreement, using five other loci (URA1, URA3, NUP116, PLB3, and SAPT4), Jacques and coworkers [192] differentiated 67 *G. candidum* strains into phylogenetically distant groups, with one being composed of environmental strains, based on the MLST scheme. One of the two identified clusters of dairy strains had limited genetic diversity, whereas the other was more correlated with the environmental isolates. Based on these identified microsatellite-like markers and focusing on twelve highly polymorphic regions, Tinsley and coworkers [194] suggested a stronger separation between the cheese isolates and the environmentally-isolated strains.

Morel and coworkers [10] analyzed the genome of *G. candidum* strain CLIB 918 (=ATCC 204307) using whole genome sequencing and comparative genomics. The analysis revealed the existence of fifty-six gene families (groups of paralogs having at least three genes), most of which are unique and whose function remains unknown. Additionally, the second largest amplification contained sixteen members, the genes of which were associated with the GRE2-like family involved in ergosterol biosynthesis and regulation of *S. cerevisiae* filamentous growth [195,196]. Other detected genes encoded for permeases, such as allantoate permeases and transporters, including nicotinic acid, bile acid, and monocarboxylate. A great number of genes were involved in chitin metabolism. Moreover, the presence of genes critical for *G. candidum* growth in dairy products, such as four genes coding for carboxylesterase/type B lipase and cheese sensorial characteristic development, such as volatile sulfur compounds, were identified. Furthermore, gene families encoding enzymes responsible for the degradation of the plant cell wall were discovered. Finally, functional annotation analysis assigned the species to the Saccharomycotina subphylum. It is noteworthy that genes with increased sequence similarity within filamentous fungi (Pezizomycotina and Basidiomycota), named SRAGs (Specifically Retained Ancestral Genes), which were considered lost from the other yeasts, were detected.

Recently, whole genome sequencing (WGS) analysis combined with phenotypic assays was applied to reveal the functional potential of *G. candidum* strains [41]. For this purpose, eight *G. candidum* and three *Galactomyces* spp. strains were used, which were phenotypically characterized using the approach described by de Hood and Smith [197]. Using the two previously identified MLST schemes [192,193], and *G. candidum* CLIB 918 [10] genome as a reference, the analysis revealed a new MLST scheme, which led to the identification of fifteen sequence types (STs) and uncovered three main complexes called GeoA, GeoB, and GeoC. In total, the genome assembly and annotation of twenty-four genomes was submitted to the Genbank database. A list of the *G. candidum* strains included in this review and their potential industrial application is shown in Table 1. More details about the identified enzymes, their UniProtKB entry, EC number, sequence length, and preliminary catalytic activities are provided in Appendix A.

## 8. Association with Diseases

### 8.1. Humans

*G. candidum* has rarely been shown to cause infection in humans. The reported cases were lower than 100 between 1842 and 2006 [2]. Cases of geotrichosis mainly involve immunosuppressive patients [198,199], patients with neutropenia, and following hematological malignancy or cytotoxic chemotherapy [200]. Some cases involve patients with renal fungal bezoar [201], acute myelogenous leukemia following allogeneic stem cell transplantation [202], hepatoblastoma [203], HIV [204], and infection after renal transplantation [205]. Sfakianakis et al. reported invasive cutaneous infection in a patient with diabetes mellitus [206]. Furthermore, a case of geotrichosis was reported in a burn patient [207] and a patient with a traumatic joint [208]. Other cases involve patients with severe illness, alcohol abuse, and chronic kidney or lung disease [209,210]. The recommended treatment for geotrichosis involves amphotericin B (alone or in combination with flucytosine) or voriconazole alone [207]. More studies need to be performed to better understand the basis of infection in these individuals and to elucidate the genes that might be associated with the geotrichosis caused by some strains. This is particularly important in light of the potential for development of *G. candidum* as an effective probiotic.

### 8.2. Animals

*G. candidum* infections are also rare in animals. However, Figueredo and coworkers [211] mentioned that *G. candidum* was responsible for 28.1% of suspected fungal infections in horses. The symptoms included, in most cases, erythematous, dry, circular alopecia and, in lower numbers, pruritus and desquamation. In agreement with these findings, Padalino and coworkers [212] reported a case where *G. candidum* caused dermatitis in a horse that had previously had surgery, antibiotic treatment, and corticosteroid treatment. The authors suggested that long-term antibiotics or corticosteroids may lead to dermatitis due to *G. candidum*. Cases of geotrichosis have been reported in other animals, such as cattle, dogs, primates, birds, turtles, and snakes [211]. In contrast to humans, in whom most cases of geotrichosis affect the respiratory system, *G. candidum* in animals affects the skin and the gastrointestinal tract.

### 8.3. Plants

Some strains of *G. candidum* are considered a critical factor in the spoilage of fruits and vegetables, such as cucumber, carrot, tomato, and stone fruit [213,214,215]. Additionally, *G. candidum* Link ex Pers. is considered responsible for citrus sour rot leading to great economic losses for citrus fruit producers [216,217]. Additionally, *G. candidum* strains were found to cause postharvest sour rot in peaches in Pakistan [218] and kiwifruits in China [219]. The most commonly used method for the control of postharvest disease in fruits and vegetables is the use of fungicides. The generally applied fungicides for this purpose include imazalil, thiabendazole, fludioxonil, sodium orthophenyl phenol, pyrimethanil, guazatine, or a combination of these compounds [220]. (However, the use of these products is becoming progressively restricted as a result of their increased residual toxicity, potential carcinogenicity, and environmental contamination [221]. Additionally, their constant use leads to the development of fungicide-resistant fungi. The application of essential oils and their primary components has been proposed as an alternative, environmentally friendly approach for the control of decay caused by *Geotrichum* species [222,223]. Mousa and coworkers [224], evaluated the effect of thymol, which is a basic component of oregano, against orange fruit infected experimentally with chemical-fungicide-resistant spores of *G. candidum* and *Penicillium digitatum*. Specifically, the *G. candidum* strain was resistant to guazatine. The study revealed that treatment with 1.0 mg/mL of thymol before inoculation with a mixture of *P. digitatum* and *G. candidum* prevented the appearance of rot. Recently, Cai, Xu and coworkers [225] proposed the use of menthol, a cyclic terpene alcohol, which is the main component of peppermint essential oil, for the control of decay caused by *Geotrichum citri-aurantii* and closely related species, such as *G. candidum*. Menthol can interact with the fungal membranes, causing a disruption in their integrity, increasing membrane permeability and releasing ROS, leading to degradation of the plasma membrane. Bourhou and coworkers [226] evaluated the sensitivity of *G. candidum* to bis(arylidene-2-[(tetrazol-5-yl)]alkane derivatives to reveal that the tested strain was sensitive at the concentration of 1 mg/mL. The strain was also sensitive to cycloheximide (1 mg/mL), but not to gentamicin (1 mg/mL).

## 9. Conclusions

The capacity of *G. candidum* to produce a great variety of biotechnologically important enzymes and secondary metabolites with antimicrobial activity renders this species a significant candidate for use in a plethora of biotechnological applications. Several strains have GRAS status, enabling their safe use in the agrifood- and bio-industries. In addition, the ability of *G. candidum* to remove/reduce pollutants, such as EEW and heavy metals from the aquatic environment, and degrade several types of organic waste from wastewater offers environmentally friendly solutions to reduce ever-increasing pollution. The information presented in this review can be used as a benchmark for future studies into the discovery of more applications involving *G. candidum*. A possible example could be the development of cheese analogs, as in the case of flaxseed oil cake for vegan and lactose-intolerant individuals. Additionally, organic biomass from *G. candidum* could be used as bionanocomposites for drug or natural component delivery purposes, as in the case of curcumin, or as antimicrobial agent bionanocomposites, possibly for skin disease treatment.

## Figures and Tables

**Table 1 foods-12-01124-t001:** List of the assembled and annotated and/or 16S rRNA gene-sequenced *G. candidum* strains, the genome size, and possible industrial application.

G. candidum Strain	Biosample	Bioproject	Assembly	Size (Mb)	Biological Process
CLIB 918	SAMEA3158493	PRJEB5752	GCA_001402995.1	24.84	1. Lipases, proteases, and volatile sulfur compound production responsible for specific aromas in cheese 2. Expression of delta 12 fatty acid desaturase for linoleic acid and α-linolenic acid production 3. Lytic polysaccharide monooxygenases production. 4. Ergosterol biosynthesis.
M2404	SAMN27964038	PRJNA833221	GCA_023629735.1	26.02	N/A
K2	SAMN20845018	PRJNA755838	GCA_020466465.1	24.98	N/A
M2401	SAMN27964034	PRJNA833221	GCA_023629795.1	26.47	N/A
Z7-1	SAMN20845971	PRJNA755888	GCA_020466515.1	25.3	N/A
QAUGC01	SAMN10962553	PRJNA523005	GCA_019450175.1	23.41	1. Administration as a probiotic strain in rohu fish, *Labeo rohita*. 2. Alkaline serine protease production, with numerous industrial applications, such as dairy food formulations.
F2203	SAMN27964033	PRJNA833221	GCA_025504395.1	23.33	N/A
PX1908B	SAMN27964030	PRJNA833221	GCA_023627875.1	23.37	N/A
Y1111	SAMN27964035	PRJNA833221	GCA_025504415.1	23.39	N/A
M1113A	SAMN27964031	PRJNA833221	GCA_025504455.1	23.38	N/A
B1101	SAMN13174491	PRJNA587010	GCA_011420275.1	23.39	N/A
B1101	SAMN27964032	PRJNA833221	GCA_025504425.1	23.39	N/A
B1109	SAMN27964037	PRJNA833221	GCA_023627915.1	23.41	N/A
LMA-1028	SAMN09709725	PRJNA482619	GCA_013365065.1	24.47	Volatile compound production that contributes to cheese sensorial characteristics development.
B1112B	SAMN27964036	PRJNA833221	GCA_025504335.1	23.41	N/A
LMA-563	SAMN09692668	PRJNA481060	GCA_025234805.1	24.7	Volatile compound production that contributes to cheese sensorial characteristics development.
LMA-40	SAMN09709337	PRJNA482576	GCA_014596635.1	24.37	Volatile compound production that contributes to cheese sensorial characteristics development.
LMA-77	SAMN09709489	PRJNA482610	GCA_013365075.1	24.02	Volatile compound production that contributes to cheese sensorial characteristics development.
Z8-4	SAMN20846412	PRJNA755896	GCA_020466505.1	24.89	N/A
LMA-317	SAMN09709581	PRJNA482616	GCA_013305595.1	24.31	Volatile compound production that contributes to cheese sensorial characteristics development.
M4009	SAMN27964039	PRJNA833221	GCA_025504325.1	23.32	N/A
LMA-70	SAMN09709440	PRJNA482605	GCA_013365125.1	24.72	Volatile compound production that contributes to cheese sensorial characteristics development.
LMA-1147	SAMN09862943	PRJNA486748	GCA_013305805.1	22.81	Volatile compound production that contributes to cheese sensorial characteristics development.
LMA-244	SAMN09709542	PRJNA482613	GCA_013365045.1	23.66	Volatile compound production that contributes to cheese sensorial characteristics development.
OMON-1, GenBank: MF431584.1				1063 bp	Antibacterial properties
Commercial strain GEO^®^	-	-	-	-	Starter culture for the production of flaxseed oil cake, an analog of Camembert cheese.
X16010211, IFBM Malting Yeast^®^	-	-	-	-	Used as biocontrol agent on T-2 toxin produced by *Fusarium sporotrichioides* and *F. langsethiae* during the malting process, phenyllactic acid production.
PF005	-	-	-	-	Alcohol acetyltransferase, GcAAT, ethyl 3-methylbutanoate, 3-methyl-1-butanol, 2-phenylethanol, naphthalene, isopentyl acetate, and isobutyl acetate production, responsible for anti-fungal activities.
HBCICC71016	-	-	-	-	Dietary supplementation for gibel carp CAS Ⅲ (*Carassius auratus gibelio*) growth.
LG-8, GenBank: MK640636.1	-	-	-	352 bp	1. *Pseudomonas aeruginosa* PAO1 has adhesion properties to *G. candidum* cells. 2. Bioremediation of Pb^2+^. 3. Novel carrier for curcumin encapsulation.
GenBank: MK880487	-	-	-	348 bp	Biosurfactant production responsible for antifungal properties.
Gad1, GenBank: MN638741	-	-	-	696 bp	Cellulase production.
3C, GenBank: KJ958925.1	-	-	-	1605 bp	Cellulase production, involved in industrial cellulase cocktails that are employed in biomass conversion and filter paper and cotton degradation.
Dec_1	-	-	-	-	Cellulase production, involved in industrial cellulase cocktails that are employed in biomass conversion and filter paper and cotton degradation. Various dye decolorization.
NRRL Y-552, undação André Tosello (Campinas, SP, Brazil)	-	-	-	-	Lipases production, decyl oleate production.
4013	-	-	-	-	Lipase production.
ATCC 34614	-	-	-	-	Lipase production.
AA15	-	-	-	-	Pectinase production for fruit juice clarification.
NBRC 4597	-	-	-	-	Acetophenone reductase production, an enzyme with broad-spectrum activity regarding the oxidation of aldehydes to carboxylic acids and selective activity for the oxidation of dialdehydes to aldehydic acids.
S12 (CCTCC AF2012005), (Wuhan, China)	-	-	-	-	Glutamate dehydrogenase production, highly active against glutamate, hexanol, α-ketoglutarate, and isoamyl alcohol.
CCT 1205, André Tosello Foundation, Parque Taquaral, Campinas-SP, belonging to the Tropical Culture Collection	-	-	-	-	ε-caprolactone production.
NBT 1, GenBank: MF461333	-	-	-	555 bp	Medium-chain fatty-acid-rich oil production.
CS-67	-	-	-	-	Bioremediation of Cu^2+^, Zn^2+^, and Ni^2+^.

## Data Availability

The data presented in this study are available on request from the corresponding author.

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
