# Peer review of "Uncovering the Biotechnological Importance of Geotrichum candidum"

_foods, 2023, doi:10.3390/foods12061124_

Round 1
Reviewer 1 Report
The article entitled of uncovering the biotechnological importance of Geotrichum candidum related to the application of the strian in verious aspects that might be not deep into foods, nutritions and related to food processing. The themes of this manuscript was not good in investigation and also did not offer any development of knowledge and trends. A novelty of this manuscript was questionable. This version of the manuscript was not suitable and needed to be improved.
Author Response
We would like to thank the reviewer for reviewing our manuscript. However, we respectfully disagree with the reviewer regarding the appropriateness of the themes covered and the novelty of the review. We agree with the other reviewers in recognizing the importance of a systems perspective on the food industry, with the broad biotechnological potential of G. candidum being exciting and highly relevant in this context. This includes aspects such as industrial enzymes, safe natural or controlled microbial cultures, antimicrobial compounds, food preservation, food waste, foodborne diseases and other applications detailed in this review. We strongly believe that this species has several important biotechnological applications of relevance to the food industry, but we respect the reviewer’s opinion to disagree with us.

Reviewer 2 Report
This review focuses on yeast Geotrichum candidum as potential commercial strain to be developed in various biotechnological applications by summarizing literature reports from year 2017 to 2022. The manuscript is in general well-written. Following are the suggestions to this manuscript:
1) The manuscript discusses about the biotechnological potential of Geotrichum candidum in relation to food preserve and detoxification. A majority of the manuscript centered around the pratical application of this yeast strain and some enzymes in relation to food preserve and quality. To better demonstrate the utility of enzymes in food preservation, a table is needed to include the name of enzyme, its sequence length, and preliminary catalytic activities. Similarly, a summary over the use of nanoparticles to remove toxic chemicals from wastewater (6.2-6.6) is needed.
2) The discussion over the genomic infomation is not sufficient and should include a table with the isolation of strain, the total genome length and its application in industrial perspectives.
3) The conclusion should bring more information relating to future application perspectives of this yeast strain.
Author Response
We would like to thank the reviewer for reviewing our manuscript and for providing us with comments that will improve the quality of our work. Please find the corrections in the manuscript using track changes.
A table that includes enzymes associated with lytic polysaccharide monooxygenases, lipases, proteases, dehydrogenases, oxidases, reductases and kinases in G. candidum genome, their UniProtKB entry and EC number, sequence length and preliminary catalytic activities was created (please find the table below). However, we would suggest this table to be added in the supplementary material. Also, a summary over the use of nanoparticles to remove toxic chemicals from wastewater was added.
The following table was created and added:
Table 1. List of the assembled and annotated and/ or 16S rRNA gene sequenced G. candidum strains, the genome size and possible industrial application.

Author Response
We would like to thank the reviewer for reviewing our manuscript and for acknowledging the quality of our work. Please find the corrections in the manuscript using track changes.
Specific comments
I would maybe change the order of paragraphs – firstly lines 45-55 and then continue with lines 37-44.
Corrected
Degradation of biogenic amines should be mentioned.
Added in the section 6.7. However, searching in NCBI using the keywords “Degradation of biogenic amines and Geotrichum candidum” identified 41 articles, from which one from 1998 mentioned the ability of Geotrichum candidum to degrade biogenic amines and two from 2002 and 2006 mentioned the ability of Geotrichum candidum to produce histamine.
Also, the ability to form biofilm by G. candidum should be mentioned.
Mentioned in line 63
Are there also some studies dealing with resistance of G. candidum against antimicrobial compounds (plant, organic acids), antibiotics or sanitizing agents? It would be beneficial to add it in context of G. candidum use as starter culture, producer of desired compounds.
There is some information about general fungicides, but not enough about G. candidum. In the Table 1 we added a list of the assembled and annotated and/ or 16S rRNA gene sequenced G. candidum strains, the genome size and possible industrial application. When trying to search for antibiotic resistant genes in the annotated genes we couldn’t find enough information. Most identified open reading frames (ORFs) were identified as hypothetical proteins. However, a study that mentioned a G. candidum strain resistant to guazatine was added in line 611 and to gentamicin in line 622.
Line 214 – name some microorganisms that are inhibited by phenyllactic acid, besides L. monocytogenes.
A study that mentions that the produced by Geotrichum candidum phenyllactic acid inhibits Fusarium sporotrichioides and F. langsethiae growth is mentioned in line 186.
Line 230 – other fungi, e.g… .. ?
such as Fusarium oxysporum, Pseudocercospora sp., and Cercospora sp.
Added in line 239.
I would maybe change the order of sentences – firstly lines 428-431 (end with “adsorption to the cell wall.”), then line 437 (from “the ability of strain LG-8…”) to line 446, then continue with lines 422 (from “The strain G. sp. C6-67…”) to lines 427 and finally end up with lines 431-437.
We would prefer to keep the original format, because we would prefer to mention a strain that survives in and accumulate high levels of Cu²⁺, Zn²⁺, and Ni²⁺, and then mention Pb2+ in a separate paragraph.
I would maybe exclude the chapters 6.3. and 6.5. as it has secondary impact on foods, food industry. Maybe it will be more suitable to just mention this ability in few sentences in conclusion or in very short chapter “X.Y. other G. candidum ability”.
We would prefer to keep them, as they indicate some additional biotechnological applications of G. candidum.
The ability of G. candidum to cause disease in animal should be added.
Added in lines 595-605.

Reviewer 4 Report
I have reviewed the review entitled Uncovering the biotechnological importance of Geotrichum candidum.
My only concern is about the title, when the name of any organism is included in the title, it is recommended that the scientific name be followed by the order and family placement.
The authors have done a thorough and comprehensive analysis of all available data on G. candidum. All information is relevant and very well analyzed and written.
Author Response
We would like to thank the reviewer for reviewing our manuscript. We appreciate the acknowledgement of the quality of our work. Since we didn't perform characterization of a new species, we can use only the species name in the title.
Round 2
Reviewer 2 Report
The revised manuscript provided summary of G. candidum genome whose major natural products include fatty acids, lipases, proteases and volatile compounds. No additional information is required.